# Resilience of Neural Cellularity to the Influence of Low Educational Level

**DOI:** 10.3390/brainsci13010104

**Published:** 2023-01-05

**Authors:** Viviane A. Carvalho de Morais, Ana V. de Oliveira-Pinto, Arthur F. Mello Neto, Jaqueline S. Freitas, Magnólia M. da Silva, Claudia Kimie Suemoto, Renata P. Leite, Lea T. Grinberg, Wilson Jacob-Filho, Carlos Pasqualucci, Ricardo Nitrini, Paulo Caramelli, Roberto Lent

**Affiliations:** 1Neuroplasticity Laboratory, Instituto de Ciências Biomédicas, Universidade Federal do Rio de Janeiro, Rio de Janeiro 21941-902, RJ, Brazil; 2Biobank for Aging Studies, Faculdade de Medicina, Universidade de São Paulo, São Paulo 01246-903, SP, Brazil; 3Memory and Aging Center, University of California San Francisco, San Francisco, CA 94158, USA; 4Laboratory of Medical Research in Aging (LIM-66), Faculdade de Medicina, Universidade de São Paulo, São Paulo 01246-903, SP, Brazil; 5Behavioral and Cognitive Neurology Research Group, Faculdade de Medicina, Universidade Federal de Minas Gerais, Belo Horizonte 30130-100, MG, Brazil; 6D’Or Institute of Research and Education, Rio de Janeiro 22281-100, RJ, Brazil

**Keywords:** literacy, medial temporal lobe, cellularity, isotropic fractionator

## Abstract

Background: Education is believed to contribute positively to brain structure and function, as well as to cognitive reserve. One of the brain regions most impacted by education is the medial temporal lobe (MTL), a region that houses the hippocampus, which has an important role in learning processes and in consolidation of memories, and is also known to undergo neurogenesis in adulthood. We aimed to investigate the influence of education on the absolute cell numbers of the MTL (comprised by the hippocampal formation, amygdala, and parahippocampal gyrus) of men without cognitive impairment. Methods: The Isotropic Fractionator technique was used to allow the anisotropic brain tissue to be transformed into an isotropic suspension of nuclei, and therefore assess the absolute cell composition of the MTL. We dissected twenty-six brains from men aged 47 to 64 years, with either low or high education. Results: A significant difference between groups was observed in brain mass, but not in MTL mass. No significant difference was found between groups in the number of total cells, number of neurons, and number of non-neuronal cells. Regression analysis showed that the total number of cells, number of neurons, and number of non-neuronal cells in MTL were not affected by education. Conclusions: The results indicate a resilience of the absolute cellular composition of the MTL of typical men to low schooling, suggesting that the cellularity of brain regions is not affected by formal education.

## 1. Introduction

Cognitive Reserve (CR) can be defined as the brain’s ability to withstand damage from a pathological condition, or simply as a result of the aging process, thus delaying the clinical manifestations of a pathology [1]. Each individual has his/her specific CR; therefore, differences can be observed between individuals regarding neural networks or cognitive processes [2], which in turn can explain why each person responds differently in the presence of a disease [1,3].

The concept of CR was proposed to elucidate the lack of a systematic relationship between the amount of brain damage and the intensity of symptoms [4]. When comparing people with the same brain alterations caused by a neuropsychiatric condition such as dementia, those with higher CR may have less severe symptoms than those with low reserve [3]. Several factors can directly impact CR, including educational level, leisure activities, and occupation, and thus can contribute to a greater tolerance to pathological events [5,6,7,8,9].

Some studies have shown an association between education and features of several cortical regions, bilaterally: premotor area, cingulate cortex, perisylvian area, and insular cortex [10]. These findings are in line with other studies that revealed a greater volume of gray matter in the temporoparietal and frontal lobes bilaterally [11], transverse temporal cortex, insula, cingulate cortex [12], right superior temporal cortex, left insula and bilateral anterior cingulate [13], left anterior cingulate cortex [14] and right middle frontal gyrus, right middle and posterior cingulate gyrus, and right inferior parietal lobe [4]. Our work focused on the medial temporal lobe (MTL).

The relevance of MTL [15] in the context of education can be emphasized by the number of recent reviews relating some of its component regions to memory [16], and therefore learning [17]. Indeed, the hippocampus and parahippocampal gyrus, in particular, have been focused on by many researchers at different levels of inquiry, from behavioral, physiological, and morphological angles, employing microscale in vitro investigations [18], in vivo approaches in animals [19,20], structural imaging [21], and postmortem morphological [22] approaches, and functional imaging [23], behavioral [24], and clinical analyses [25].

At the cellular level, few studies have been conducted to clarify the mechanisms involved in literacy and its neurobiological consequences. Jacobs et al. [26] demonstrated that, in cognitively healthy individuals, education exerted an important influence on the dendritic arborization of pyramidal neurons in the regions classically known as Wernicke’s area. It is also known that the greater the number of school years, the greater the performance on verbal memory tests in older adults. Furthermore, education can affect morphometric measures of hippocampal subregions, namely CA1 and the subiculum [27]. However, nothing is known about the influence of education on the number of total cells, neurons, and non-neurons in the MTL, particularly with regard to the hippocampus.

At the macroscopic level, Brayne et al. [28] found that higher education was associated with a greater brain mass and lower risk of dementia. Along the same lines, Ferretti-Rebustini et al. [29] also demonstrated that higher education was associated with greater brain mass. More specifically, in individuals with a higher education, the decline in hippocampal volume is less pronounced with age than in individuals with low education [30]. An older adult cohort study evaluated how higher education can maintain cognitive functions over a 10-year period. Preservation of cognitive function was shown to be associated with a greater volume of MTL at baseline when compared to older people with cognitive decline [31].

Education can also exert direct and indirect influences on CR, particularly in regions that are more vulnerable to dementia and Alzheimer’s disease [10]. It is believed that education can protect the brain or provide resilience against dementia [7,28]. In older people with a higher educational level with Alzheimer’s dementia, better cognitive performance was observed when compared to those with Alzheimer’s dementia and less education [32,33], suggesting an important role of education in maintaining brain function, even when there is a pathological process installed [34].

Formal education consists of the systematic transmission and acquisition of new information and skills for better social and intellectual performance, as well as professional training. It encompasses the continuous improvement of various cognitive functions such as memory, language, reasoning, and socioemotional skills, and usually takes place throughout childhood and early adulthood, when the brain is more plastic [35]. Therefore, it is conceivable that this intensive acquisition of new functions and skills leads to changes and adaptations in several brain areas [36].

The level of education can affect an individual’s general health, as more educated people can generally have higher life expectancy compared to less educated subjects [37,38,39]. Recent discoveries using neuroimaging techniques and neuropsychological assessments shed light on the ability of the human brain to acquire new cultural skills, such as reading and arithmetic [40]. Consistent evidence indicates that reading acquisition is a milestone in an individual’s development and is accompanied by structural changes in the brain both of individuals who learned to read during childhood [41] and in those who learned to read later, during adulthood [42].

Not only high educational levels promote better performance of executive functions; even a few years of schooling are enough to promote changes in brain structure and function [43,44,45]. Individuals with an average of four years of formal education, compared to illiterate individuals, had higher fractional anisotropy values in the upper right longitudinal fasciculus, suggesting that even a few years of formal education can contribute not only to CR, but also to greater integrity of the white matter microstructure in regions that connect the visual cortex with language areas [46].

As mentioned before, we tackled this issue by focusing on MTL, due to its role in memory, and therefore learning and education. To the best of our knowledge, no study has compared the MTL mass between individuals with high and low education. In addition, no study has evaluated whether education affects the absolute cellular composition of the brain, especially the MTL.

The donation and storage of post-mortem brain tissue in brain banks have driven research towards many discoveries, improving our understanding of the functioning of the nervous system [47,48,49]. Post-mortem brain studies can be performed employing stereological analyses on histologically processed tissues [50,51,52,53] or by using the Isotropic Fractionator (IF) method [54,55,56]. IF is a technique created by Herculano-Houzel and Lent [57], which consists of transforming anisotropic brain tissue into an isotropic suspension of nuclei, labeled with specific and universal molecular markers. Investigation in post-mortem human brains using IF has made important discoveries and has broken some dogmas [58]. Building on these previous works, the objective of this study was to investigate the influence of education on the absolute quantitative cellular composition of the MTL (comprising the hippocampus, amygdala, and parahippocampal gyrus) of men without cognitive impairment. The work addresses the possible influence of cultural variables on the cellular underpinnings of CR.

## 2. Materials and Methods

### 2.1. Subjects

This is a cross-sectional study approved by the Research Ethics Committees of the D’Or Institute and of the University of São Paulo Medical School. A total of 26 individuals were studied, with brains provided by the Biobank for Aging Studies of the University of São Paulo, Brazil. Once the primary relatives or caregivers agreed to donate the brains, an informed consent form was signed.

### 2.2. Inclusion and Exclusion Criteria

The subjects included in the study were middle-aged men between 47 and 64 years, with low formal education (0–4 school years) or high formal education (≥8 school years). Cases with intermediate periods of schooling (5–7 years) were not included. Choice of these ages was made to guarantee the interruption of formal education in youth, and to avoid the incidence of cognitive decline in the elderly. The informant needed to be capable of adequately responding to the clinical interview. Cases with congenital malformations, disseminated cancer, stroke, drug abuse, alcohol-use disorder, mental illness, and dementia or mild cognitive impairment were excluded. A detailed histopathological examination was used to exclude brains with signs of lesions, neurodegenerative diseases, tumors, hippocampal sclerosis, and other pathologies.

Clinical interviews were carried out with the subjects’ relatives to obtain sociodemographic data, clinical background, and lifestyle habits. Retrospective cognitive and functional assessments were obtained through the Clinical Dementia Rating (CDR) [59] and the Informant Questionnaire on Cognitive Decline in the Elderly (IQCODE) [60]. Only cognitively normal subjects (CDR = 0 and IQCODE ≤ 3.4) were included. Men only, not women, were chosen to avoid sex dimorphisms, and age range was defined to avoid brain changes with aging, as previously shown [61].

### 2.3. Dissection of the MTL

After brain removal, each left hemisphere was fixed in 2% paraformaldehyde (PFA) for 36–40 h. Then, the material was moved to phosphate-buffered saline (PBS) and maintained in the refrigerator at 4 °C until processing. First, meninges and blood vessels were removed. Subsequently, the medial temporal lobe was dissected employing a standardized anatomical protocol as used by Oliveira-Pinto et al. [61].

### 2.4. Chemomecanical Dissociation

After dissecting the MTL, the tissue was weighted and sliced into smaller fragments. Then, a chemomechanical dissociation was performed, which consisted of tissue maceration in a dissociation solution composed of 40 mM sodium citrate and 1% Triton X-100. The tissue fragments were placed in potters coupled with a semiautomatic machine with moving pistons controlled for translation and rotation with the help of a computer [62]. This procedure made it possible to rupture the plasma membrane of the cells, leaving the nuclear membrane intact, thus allowing the subsequent counting of the cell nuclei. In the nervous tissue, since every cell contains only one nucleus, and the nuclear suspension is isotropic, the number found is an accurate estimate of the total number of MTL cells [57,63].

### 2.5. Immunocytochemistry

To quantify the total number of cells, all nuclei were labeled with 2% DAPI (4′-6-diamino-2-phenylindol dihydrochloride), a DNA-specific marker. Aliquots of 10 µL from the isotropic suspension were collected and placed in a hemocytometer (Neubauer chamber) for counting using a fluorescence microscope (Zeiss AxioImage). When the counts varied by more than 15%, the suspension was not considered isotropic and further fractionation was performed before resuming counting. Only intact, DAPI-stained nuclei, or those slightly damaged by fractionation, were counted.

Once the number of total cells was obtained, a 1000 µL aliquot of the nuclei suspension was employed for immunocytochemical labeling of the NeuN protein. Samples were subsequently centrifuged for five minutes, washed three times with 1000 µL PBS, and incubated overnight in primary anti-NeuN IgG antibody produced in mice (1:200, 200 µL of PBS). The next day, the samples were centrifuged for five minutes and washed three times with PBS, incubated for two hours in Alexa Fluor 546 secondary antibody (1:300, 260 µL of PBS, 30 µL of normal goat serum, and 10 µL of DAPI), centrifuged for five minutes, washed three times with PBS, and resuspended in PBS. Then, 10 µL aliquots were removed from the samples and placed in the Neubauer chamber for counting under the fluorescence microscope (Zeiss AxioImage).

Of a total of 500 DAPI-stained cell nuclei from each sample, those colocalized with NeuN were counted. Therefore, it was possible to obtain the number of total cells (density of DAPI+ nuclei in the suspension multiplied by the total volume of the suspension), the number of neurons (total number of DAPI+ nuclei multiplied by the percentage of those doubly labeled with DAPI and NeuN), and that of non-neuronal cells (estimated by subtraction of the number of neuronal nuclei from the total number of nuclei in the suspension). The entire experimental procedure was blindly conducted regarding the subjects’ data. The MTL mass and number of neurons, number of non-neurons, and total number of cells found in the left hemisphere were multiplied by 2 to represent both hemispheres.

### 2.6. Statistical Analysis

Statistical analysis was conducted using STATA software (version 13). A significance level of 5% (α = 0.05) was considered. Data for sociodemographic variables were analyzed and presented as mean and standard deviation. For the comparison of brain mass, MTL mass, and number of cells between the high and low education groups, Student’s Independent Test was used. The mean and standard deviation graphs with the number of neurons, non-neuronal cells, and total cells were generated in GraphPad Prism, version 5.0.

Unadjusted linear regression tests were used to investigate the association between education and the absolute cellular composition of MTL. Subsequently, we adjusted these models for age, height, and Braak stage for neurofibrillary distribution to account for possible confounding factors. We also performed unadjusted linear regression tests to investigate the association of education as a continuous measure, and the absolute cellular composition of MTL.

## 3. Results

### 3.1. Clinical and Demographic Data of Participants with Collected Brain Samples

The sociodemographic data of the subjects are shown in Table 1. The mean age of the sample was 56.7 (±5.6) years, ranging from 47 to 64 years. The mean educational level was 7.5 (±4.5) years, ranging from 0 to 16 years. All 26 subjects scored 0 on the CDR and 3 on the IQCODE.

### 3.2. Brain Mass and MTL Mass Related to High and Low Educational Level

The average brain mass in the high education group was 1386 g (±168.1), compared to 1267 g (±91.5) in the low education group. Student’s Independent Test showed that, on average, the groups with high and low education present a significant difference in brain mass (*p* = 0.044; Figure 1A).

The average MTL mass in the high education group was 31.35 g (±3.9), similar to the low education subjects with 32.10 g (±5.5). Student’s Independent Test showed that, on average, the groups with high and low education did not show a significant difference in MTL mass (*p* = 0.81; Figure 1B).

### 3.3. Cell Numbers (Neurons, Non-Neurons, and All Cells) Related to High and Low Educational Level

The average total number of MTL cells in the high education group was 1.93 (±0.28) billion, compared with 1.95 (±0.33) billion among the low education subjects. Student’s Independent Test showed that the two groups did not present a significant difference in the total number of cells (*p* = 0.93; Figure 2).

The average number of neurons in the high education group was 0.30 (±0.17) billion, against 0.39 (±0.23) billion neurons in the low education group. Student’s Independent Test showed that the groups with high and low schooling did not show a significant difference in number of neurons (*p* = 0.69; Figure 2).

The average number of non-neurons among high education subjects was 1.63 (±0.37) billion cells, compared with an average of 1.56 (±0.33) billion cells in the low education group. Student’s Independent Test showed that there was no significant difference in number of non-neurons cells (*p* = 0.31; Figure 2).

Unadjusted linear regression tests showed that education was not associated with the total number of cells, the number of neurons, nor the number of non-neurons, indicating that the cellularity of MTL was not affected by educational level (Table 2). Additionally, unadjusted linear regression tests taking education as a continuous measure showed it was not associated with the same variables, confirming that cellularity of MTL was not affected by schooling (Table 3).

## 4. Discussion

This work has investigated the possible impact of education on the microscale structure of the MTL in humans. Despite reduced brain mass in subjects with low education, no difference was found in neuronal and non-neuronal absolute numbers within the MTL in relation to education.

Several studies have addressed the importance of education for cognitive function [11,31] and its ability to promote changes in brain structure [10,37,42,64,65,66]. However, the mechanisms involved in these changes and their potential benefits still need to be better clarified. It is conceivable, therefore, that education may be an environmental factor that contributes to cognitive and brain reserve, explainable by either the size of the brain, the number of neurons, or the number of synapses [2]. Our work confirms the findings of two previous studies that observed a greater brain mass in individuals with high education than in those with low education [28,29]. Mass and volume measured for the whole brain, however, may conceal localized changes, and do not reveal which microscale elements are responsible for the identified alterations: Neurons? Glial cells? Synapses? Where exactly in the whole brain? Furthermore, if higher educational levels do improve or protect the brain, would the opposite contribute to cognitive decline? How resilient are brain units—neurons, especially—in relation to low education?

The results of the present study suggest that the level of education is not capable of causing changes in the absolute cellularity of the MTL, indicating that cells in this brain region are resilient to this environmental influence. Using the isotropic fractionator, we verified that education did not affect the number of total cells, neurons, and non-neuronal cells in the MTL of cognitively healthy men. Thus, the absolute cellular composition of MTL may not be involved with the structural changes observed in magnetic resonance imaging studies [10,65,67]. Furthermore, the number of cells in the MTL also does not explain the better cognitive performance observed in individuals with high education [11,31,68].

The absence of a significant association between education and the absolute cellular composition of the MTL can also be explained by the operational definition of high and low education used in this study, since the low level (i.e., four years of schooling or less) could already be enough to explain the resilience of cellularity to change as a result of this variable [43,44,45]. Another possible explanation for the absence of associations between cell counts and education is the small sample size, mainly for associations between education and the total cells and neurons. The post hoc power results for these cells were 43% and 29%, raising the possibility of a type 2 error. On the other hand, the estimated power was 78% for non-neurons, indicating that the absence of association between these cells and education is quite acceptable.

The study by Resende et al. [44] in a sample of older adults with four years of formal education demonstrated a strong association between left hippocampal volume and episodic memory scores. Therefore, even very few years of study might be enough to modify the brain’s macrostructure and thus build CR. However, as shown by the present data, this increase in volume would not be explained by an increase in cellularity.

Consequently, other mechanisms may underpin the benefits caused by education, namely angiogenesis, synaptogenesis, or myelination [2], and not necessarily neurogenesis or programmed cell death, phenomena that impact cellularity. Recently, there has been much debate about the existence of neurogenesis in adulthood, and there is still controversy as to whether the production of new neurons indeed takes place in the adult human brain [69] or not [70]. Neuronal recycling, a phenomenon that has been proposed for promoting changes in brain functionality, may explain why education did not affect cell composition in the MTL. This term can be defined as neuronal connections that compose existing networks, recycled to be used in a different function. It is a form of neuroplasticity that occurs when the individual acquires some cultural or cognitive skills; for example, reading [64].

Before reading acquisition, this neural network responds to images of faces and objects, but with literacy, it starts to respond to images of letters and words [64,65], while the homologous region located in the right hemisphere continues to respond to faces [42]. Conceivably, the neuronal recycling taking place in the visual word form area (VWFA) could also occur in other brain areas, thus reorganizing preexisting circuits, which will be led to play new roles.

As well as reading, mathematics is also an important culturally acquired cognitive skill. Functional magnetic resonance imaging studies have found that some regions of the MTL participate in numerical learning [71,72,73]. Other studies have demonstrated the electrophysiological activity of neurons in the MTL during the performance of a calculation task through microelectrodes implanted in the nervous tissue [74,75]. Therefore, it is conceivable that neuronal recycling also takes place in the MTL, more precisely in the hippocampus, just as it does in the VWFA.

In addition to the neuronal recycling discussed above, white matter properties are also associated with reading ability [76]. Synaptogenesis and myelination are two mechanisms involved in plasticity and learning processes [77]. Studies have revealed changes in the white matter of the human brain after learning [45,76,77]. Indeed, differences were observed in the fractional anisotropy of the left inferior and arcuate longitudinal fascicles of children with above-average reading ability compared to children with below-average reading ability [76]. In addition to children, older individuals with an average of four years of formal education, when compared to illiterate individuals, had higher fractional anisotropy values in the right upper longitudinal fasciculus [45]. Several histological features can affect white matter anisotropy, including the number and thickness of axons [78]. In addition, it could be explained by the myelination of nerve bundles [79]. As a result, the difference observed in cognitive performance between more educated and less educated individuals [68], and between illiterate individuals and those with low education [45], may be due to a difference in the conduction speed of neural circuits, as this property optimizes the transmission time of information conveyed by the neural circuits [77].

Our study has some limitations. Among them, we can mention the inclusion and exclusion criteria, which may restrict the interpretation of our results to a limited group of individuals (men without cognitive impairment, age range between 47 and 64 years). Furthermore, as this is a cross-sectional study, we cannot establish a causal relationship between education and cell numbers. The participant’s nutritional status and the practice of leisure and physical activities are other factors that can interfere with cognitive function and brain structure. However, the difficulty obtaining detailed information about these variables in our cohort did not allow the acquisition of these data. The absolute cell composition was evaluated in the MTL as a whole, a region that houses the amygdala, the parahippocampal gyrus, and the hippocampus, but it is known that one of the structures that can benefit from education is the hippocampus [30], and within it, the dentate gyrus, which presents neurogenesis well demonstrated in adulthood. However, the difficulty of performing standardized dissection protocols of these subregions made it impossible to analyze them specifically with our technique. Therefore, it could be the case that although there is a difference in the number of cells in the hippocampus, other structures—for example, the amygdala or the cortex of the parahippocampal gyrus—may have masked the effect. Another limitation is the small sample size, as mentioned above. Post hoc power calculations ranged from 29% to 78%, with estimated sample sizes from 28 to 128 individuals. However, it is important to note that these numbers are very difficult to reach for human brains in optimal conditions to undergo counts with the isotropic fractionator.

Despite the above limitations, this is the first study that evaluated the influence of education on the number of total cells, neurons, and non-neuronal cells in the MTL of individuals without cognitive impairment. These factors are already well defined for promoting benefits to brain health, albeit with poorly understood mechanisms.

## 5. Conclusions

In conclusion, the results of this study suggest that the number of total cells, the number of neurons, and the number of non-neuronal cells in the MTL of men without cognitive impairment are not affected by educational level. The absence of association in our study may indicate that brain cellularity is resilient to these environmental factors. This finding may have an appreciable evolutionary significance, hindering the erosion of neural hardware, which is, moreover, irreplaceable during life due to the lack of appreciable cell proliferation in most neural regions. Therefore, we conclude that the differences observed in the brains of illiterate and literate individuals, as revealed by the various studies mentioned above, cannot be explained by alterations in the cellularity of the MTL, which proved resilient to this type of environmental influence.

## Figures and Tables

**Figure 1 brainsci-13-00104-f001:**
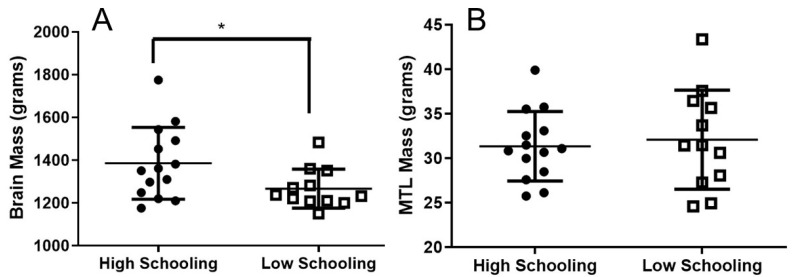
Brain mass and medial temporal lobe mass of subjects with high and low education. (**A**). Average brain mass for both groups, showing high education subjects with larger mass than those of low education. Asterisk denotes significant statistical differences between high and low education (*p* < 0.05). (**B**). MTL mass of subjects with high and low education. Differences were non-significant. (**A**,**B**). Means and standard deviations are indicated by black bars.

**Figure 2 brainsci-13-00104-f002:**
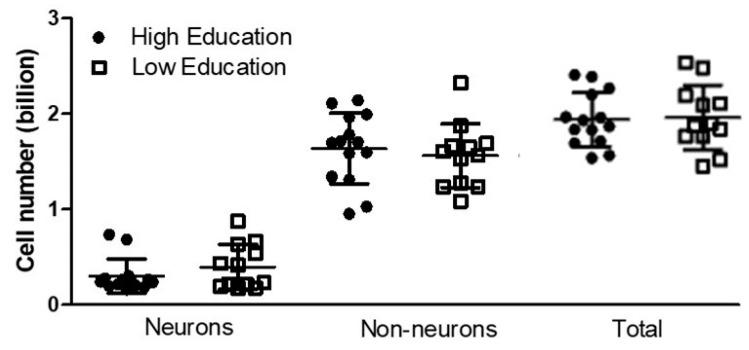
Number of cells of subjects with high and low education in the medial temporal lobe. Differences were non-significant for the number of neurons, number of non-neurons, and total number of cells between high and low education groups. Means and standard deviations are indicated by black bars.

**Table 1 brainsci-13-00104-t001:** Demographic and autopsy-related data.

Case	Age at Death	Schooling	Cause of Death	Time from Death to Fixation	Cerebral Weight (g)	MTL Weight(g)	Braak	CERAD	AP Diagnostic
1	62	8	Acute myocardial infarction	11 h 30 min	1220	31.1	2	0	Normal
2	47	8	Pulmonary edema	18 h	1582	39.9	0	0	Normal
3	49	8	Pulmonary Alveolar Hemorrhage	16 h 38 min	1210	35.76	0	0	Normal
4	63	11	Retroperitonial Hemorrhage	12 h 10 min	1176	27.6	1	0	Normal
5	56	12	Acute Pulmonary edema	11 h 20 min	1298	35.54	1	0	Normal
6	64	11	Acute myocardial infarction	18 h 42 min	1453	32.54	1	0	Normal
7	47	16	Broncho-pneumonia	13 h 30 min	1775	30.84	1	0	Normal
8	54	15	Pulmonary thromboembolism	8 h 01 min	1310	30.68	0	0	Normal
9	48	11	Acute myocardial infarction	16 h 07 min	1362	26.14	2	0	Normal
10	49	15	Pulmonary edema	14 h 59 min	1492	33.1	0	0	Normal
11	61	8	Acute Pulmonary edema	9 h 25 min	1544	29.98	1	0	Normal
12	55	12	Pulmonary thromboembolism	18 h 55 min	1249	28.48	0	0	Normal
13	58	11	Pulmonary infarction	12 h 45 min	1382	25.76	1	0	Normal
14	58	11	Pulmonary thromboembolism	14 h 52 min	1351	31.48	2	0	Normal
15	59	4	Acute renal failure	13 h 35 min	1271	43.36	2	0	Normal
16	58	4	Sepsis	15 h 25 min	1210	24.94	2	0	Normal
17	59	4	Bilateral caseous bronchopneumonia	10 h 18 min	1360	33.7	1	0	Normal
18	53	2	Pulmonary tuberculosis	13 h 24 min	1282	31.44	3	0	Normal
19	49	4	Chronic Pneumopathy	11 h	1221	31.46	1	0	Normal
20	55	4	Pulmonary edema	12 h 19 min	1150	37.58	1	1	Normal
21	62	0	Septic shock	17 h 57 min	1209	27.28	1	0	Normal
22	60	1	Acute myocardial infarction	9 h 40 min	1484	35.66	1	A	Normal
23	62	4	Pulmonary edema	13 h 38 min	1238	24.6	2	0	Normal
24	62	4	Hemiperitoneum	12 h 1 min	1232	28.08	0	0	Normal
25	64	4	Broncho-pneumonia	16 h 25 min	1200	30.62	0	0	Normal
26	62	4	Acute Pulmonary edema	19 h 26 min	1352	36.44	3	0	Normal

**Table 2 brainsci-13-00104-t002:** Association between education and absolute cell composition of the medial temporal lobe (*n* = 26).

	Unadjusted Model		Model 1 *		Model 2 **	
	β (95% CI)	*p*	β (95% CI)	*p*	β (95% CI)	*p*
**Neurons**	0.95 (−0.07; 0.26)	0.25	0.12 (−0.08; 0.32)	0.22	0.12 (−0.09; 0.34)	0.23
**Non-neurons**	−0.08 (−0.36; 0.21)	0.59	−0.17 (−0.51; 0.16)	0.30	−0.20 (−0.55; 0.15)	0.26
**Total cells**	0.20 (−0.23; 0.27)	0.87	−0.05 (−0.35; 0.24)	0.71	−0.07 (−0.38; 0.24)	0.63

* Model 1: Linear regression models adjusted for age and height. ** Model 2: Linear regression models adjusted for age, height, and Braak stage for neurofibrillary tangles.

**Table 3 brainsci-13-00104-t003:** Association between education as a continuous measure and absolute cell composition of the medial temporal lobe (*n* = 26).

	Unadjusted Model		Model 1 *		Model 2 **	
	β (95% CI)	*p*	β (95% CI)	*p*	β (95% CI)	*p*
**Neurons**	−0.003 (−0.022; 0.015)	0.70	−0.005 (−0.030; 0.020)	0.70	−0.005 (−0.030; 0.021)	0.71
**Non-neurons**	0.005 (−0.027; 0.037)	0.74	0.020 (−0.021; 0.060)	0.33	0.022 (−0.020; 0.064	0.29
**Total cells**	0.002 (−0.026; 0.029)	0.90	0.015 (−0.020; 0.050)	0.39	0.018 (−0.019; 0.054)	0.33

* Model 1: Linear regression models adjusted for age and height. ** Model 2: Linear regression models adjusted for age, height, and Braak stage for neurofibrillary tangles.

## Data Availability

Data are available from the first author upon justified request.

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
