# Peer review of "Resilience of Neural Cellularity to the Influence of Low Educational Level"

_brainsci, 2023, doi:10.3390/brainsci13010104_

Round 1
Reviewer 1 Report
Dear Authors,
Below are my comments
1. The introduction need yo be precise and focus on establishing the gap in relation to medial temporal lobe.
2.In material and methods
1. The characteristics of sampling methods should be explain
2. The reason and citation of using age 40-64 years
3. Inclusion and exclusion criteria need to specified
4. what are criteria that used to defined high and low education
5. Is there any correlation between clinical interviews and rating scale such as Clinical Dementia Rating scale that may associated with education or MTL
Author Response
-The introduction needs to be precise and focus on establishing the gap in relation to medial temporal lobe.
We believe that the Introduction contains detailed information to reinforce the importance of studying the hippocampus and the medial temporal lobe, important brain region and subregion involved in learning and memory (check page 2, lines 82-88, and page 3, lines 108-112).
-In material and methods, (1) The characteristics of sampling methods should be explained; and (2) The reason and citation of using 40-64 years. (3) Inclusion and exclusion criteria need to be specified; (4) What are the criteria used to define high and low education. (5) Is there any correlation between clinical interviews and rating scale such as CDR that may be associated with education or MTL.
- Sampling methods are now commented at page 3, lines 133-149. Sample size has been discussed as a limitation of the work in page 9, lines 364-368, due to the inherent difficulties of collecting human whole brains for this kind of study.
- Sex and age choices are now commented, as suggested, in page 3, lines 147-149.
- Inclusion and exclusion criteria have now been specified as a section in methods, and duely detailed as suggested (page 3, lines 133-149).
- High and low education have been now described in more detail in page 3, lines 133-135.
- Information about the subjects (CDR and IQCODE) were provided by the primary caretakers, responsible for donating the brains and signing the appropriate informed content terms. For this reason, they do not relate to the educational level of the subjects.
Reviewer 2 Report
The paper presents an empirical study investigating potential resilience of neural cellularity to the influence of low educational level.
The manuscript is well written and the results of interest.
Important strengths include the detailed investigations.
From the topic addressed, it could nicely fit into the Special Issue “Environmental Exposures, Neurodevelopment, and Mental Health”.
However, the sample is very small. Only twenty-six brains were investigated. Has an a priori power analysis been conducted? Are estimates reliable with such a small sample? The null findings discussed could be due to the small statistical power instead of true null effects. Please discuss this major limitation in more detail.
Investigations included men aged 40 to 64 years. What about women? What about old age? It seems probable that the pattern of associations may substantially differ in other groups. Especially resilience may become more evident in older age. The fact that findings are limited to middle adulthood needs to be discussed as a further limitation.
It needs to be better elaborated on the novelty of the paper. What do we learn from this study? How results advance cognitive reserve research beyond the many papers already existing?
Minor:
Maybe consider changing the title since the general effects implied in the title do not match the null findings discussed.
“brains were obtained” – surely it would be better to change the wording to “investigated” since brains may not be obtained.
Author Response
-(1)The paper presents an empirical study investigating potential resilience of neural cellularity to the influence of low educational level; (2) The manuscript is well written and the results of interest; (3) Important strengths include the detailed investigations; (4) From the topic addressed, it could nicely fit into the Special Issue “Environmental Exposures, Neurodevelopment, and Mental Health”.
We thank the referee for the praise of our work, and for the suggestion about the special issue.
-(5)However, the sample is very small. Only twenty-six brains were investigated. Has an a priori power analysis been conducted? Are estimates reliable with such a small sample? The null findings discussed could be due to the small statistical power instead of true null effects. Please discuss this major limitation in more detail.
The referee is correct in his/her criticism. We have performed a post-hoc power calculation, and inserted a comment on this issue by expanding the paragraph on limitations of the study (page 9, lines 364-368)
-(6)Investigations included men aged 40 to 64 years. What about women? What about old age? It seems probable that the pattern of associations may substantially differ in other groups. (7) Especially resilience may become more evident in older age. The fact that findings are limited to middle adulthood needs to be discussed as a further limitation.
Important issues raised by the referee. They are now better addressed in the new section on Inclusion and exclusion criteria (page 3, lines 133-149). In the Discussion, the issues are commented in pages 8-9, lines 348-353.
-(8) It needs to be better elaborated on the novelty of the paper. What do we learn from this study? How results advance cognitive reserve research beyond the many papers already existing?
We believe this general suggestion is fulfilled in the Discussion and Conclusions.
-(9) Minor:(9a) Maybe consider changing the title since the general effects implied in the title do not match the null findings discussed. (9b) “brains were obtained” – surely it would be better to change the wording to “investigated” since brains may not be obtained.
We preferred to maintain the title, with the argument that the negative results obtained do express the resilience of the brain cellularity to be impacted by some environmental influences such as formal education. This argument is detailed in the Introduction and the Discussion. As to the wording suggestion, it was accepted and corrected (“dissected”, instead of “obtained”) in the revised version (Abstract page 1, line 17).